# Management and Prognosis of Acute Post-Cataract Surgery Endophthalmitis: A 10-Year Retrospective Analysis in Eastern China

**DOI:** 10.3390/antibiotics12121670

**Published:** 2023-11-28

**Authors:** Xiuwen Zhang, Zhi Chen, Xiaoxia Li, Zimei Zhou, Maureen Boost, Taomin Huang, Xingtao Zhou

**Affiliations:** 1Department of Pharmacy, Eye & ENT Hospital, Fudan University, Shanghai 200031, China; xiuwen.zhang@fdeent.org (X.Z.); xiaoxia.li@fdeent.org (X.L.); 2Eye Institute and Department of Ophthalmology, Eye & ENT Hospital, Fudan University, Shanghai 200031, China; peter459@aliyun.com; 3NHC Key Laboratory of Myopia, Fudan University, Shanghai 200031, China; 4Key Laboratory of Myopia, Chinese Academy of Medical Sciences, Shanghai 200031, China; 5Shanghai Key Laboratory of Visual Impairment and Restoration, Shanghai 200031, China; 6Department of Ophthalmology, BronxCare Health System, Bronx, NY 10456, USA; zzhou@bronxcare.org; 7School of Optometry, The Hong Kong Polytechnic University, Hong Kong SAR 999077, China; maureen.boost@polyu.edu.hk

**Keywords:** endophthalmitis, cataract surgery, intravitreal corticosteroids, systematic antibiotics, visual outcomes

## Abstract

Acute post-cataract surgery endophthalmitis (APSE) is a serious vision-threatening complication of cataract surgery. Analysis of the management and prognosis in cases of APSE may provide better guidance for future treatment. Fifty-six patients (56 eyes) diagnosed with APSE between 2013 and 2022 were retrospectively reviewed. The incidence of APSE rate was 0.020% (95% CI: 0.011–0.029%). Intraocular cultures were positive in 18 (32.1%) cases, with 21 organisms isolated. Coagulase-negative *staphylococci* was the predominant isolate (12/21; 57.1%). The time from surgery to the onset of endophthalmitis was 7 days (interquartile range: 3–16) in patients with good best-corrected visual acuity (BCVA) (≥20/70) and 3 days (interquartile range: 1–8) in those with poor BCVA (<20/70). Multivariate linear regression analysis revealed that initial BCVA (logMAR) (*p* < 0.001), time from onset to initial intravitreal antibiotics (IVAs) (*p* < 0.001), and positive culture of highly virulent pathogens (*p* = 0.018) displayed significantly positive associations with the final BCVA (logMAR). Adjunctive use of intravitreal corticosteroids and systemic antibiotics were unrelated to a favorable final BCVA. In conclusion, the severity of the visual condition at baseline, as well as delayed treatment, are risk factors for poor visual outcomes in APSE.

## 1. Introduction

Cataracts are considered to be the most prevalent cause of visual impairment worldwide [1]. In China, the largest developing country with a population of approximately 1.4 billion, the annual rate of cataract surgery had risen to 2662 cases per million by the year 2020 [2]. Post-operative endophthalmitis, defined as severe inflammation involving both the anterior and posterior segments of the eye following intraocular surgery, is a serious vision-threatening complication of cataract surgery and remains a major concern for both ophthalmic surgeons and patients [3]. Several prophylactic approaches have been deployed, including pre-operative disinfection with the povidone prophylactic, local use of antibiotic drops, and anterior chamber injection of antibiotics at the conclusion of the operation [4,5,6,7]. Despite these efforts, the reported incidence of acute post-cataract surgery endophthalmitis (APSE) ranges from 0.03% to 0.15% worldwide [2]. Obviously, there is still a wide disparity in the levels of healthcare provided in different regions and hospitals due to differences in social and economic factors in reporting countries, resulting in a wide variation in the incidence of APSE [8,9].

The diagnosis of endophthalmitis is a medical emergency and needs to be addressed immediately. Pars plana vitrectomy (PPV) accompanied with intravitreal antibiotics (IVAs) have been considered the reference standard treatment for bacterial endophthalmitis [10,11,12]. However, the adjunct use of intravitreal corticosteroids and/or systemic antibiotics in the treatment of patients with endophthalmitis has been controversial [13,14]. Furthermore, there are no clear guidelines on the adjunctive management of acute bacterial endophthalmitis or clinical evidence of their effectiveness.

This study provided an overview of the incidence, clinical presentation, visual outcomes, microbiological features, and management methods of all APSE cases in a large university teaching hospital in eastern China over the past ten years. In addition, this study also aimed to explore the effectiveness of adjunctive treatments for APSE and to evaluate potential predictors for favorable visual outcomes.

## 2. Results

### 2.1. Incidence and Clinical Characteristics

Our chart review confirmed that there were 56 patients (56 eyes) who were ultimately diagnosed with APSE and met the study inclusion criteria (Figure 1). Twenty-nine of these patients underwent cataract surgery at our hospital, while the other 27 had undergone cataract surgery at outside hospitals. As a total of 153,432 cataract surgeries were performed in our hospital over the past ten years, the overall APSE rate was 29/153,432 (0.020%; 95% CI: 0.011–0.029%).

Table 1 summarizes the baseline characteristics of patients regarding the final best-corrected visual acuity (BCVA). Briefly, of 56 eyes, 44 (78.6%) were initially treated with PPV + IVAs, and 12 (21.4%) eyes were treated with IVAs alone. Of these, six (13.6%, 6/44) and six (50.0%, 6/12) eyes received additional PPV and/or IVAs treatment, respectively. At the onset of endophthalmitis, the median initial BCVA (logMAR) was 2.3 (interquartile range: 1.8–2.7). The average follow-up time was nine months (range: 1–34.8 months). The median final BCVA (logMAR) was 1.2 (interquartile range: 0.3–2.3). Twenty-one (37.5%) eyes had a final BCVA of better than 20/70, and only one eye (1.8%) was enucleated due to uncontrollable endophthalmitis.

### 2.2. Factors Affecting Visual Outcomes

In the univariate analysis, patients with final favorable vision (BCVA ≥ 20/70) exhibited better initial BCVA (logMAR), longer intervals between cataract surgery and onset of endophthalmitis, shorter time from onset to initial IVAs, and were less likely to be infected with highly virulent pathogens (Table 1). Multivariate linear regression analysis subsequently revealed that initial BCVA (logMAR) (standardized *β* = 0.407, *p* < 0.001), time from onset to initial IVAs (standardized *β* = 0.383, *p* < 0.001), and positive culture of highly virulent pathogens (standardized *β* = 0.286, *p* = 0.018) demonstrated significantly positive associations with the final BCVA (logMAR) (Table 2).

### 2.3. Culture Results

Intraocular specimens, including 48 vitreous samples and 31 aqueous samples, were collected from all 56 patients, but only 18 eyes yielded growth on culture (32.1%), with a total of 21 isolates. The rates of positive culture were 35.4% (17/48) in vitreous samples and 29.0% (9/31) in aqueous samples. Of the 21 isolates, the most frequently isolated organisms were Coagulase-negative *staphylococci* (12/21; 57.1%), followed by *Pseudomonas aeruginosa* (2/21; 9.5%), *Staphylococcus aureus* (2/21; 9.5%), and *Streptococcus species* (2/21; 9.5%) (Table 3). Infections caused by highly virulent pathogens were associated with more advanced clinical presentations, such as a shorter time to onset of symptoms after surgery and displaying symptoms of a hypopyon. The poorer visual outcomes and shorter time from onset to initial IVAs were also more frequent in patients with highly virulent pathogens (Table 4).

### 2.4. Adjunct Use of Intravitreal Corticosteroids

Overall, fifteen patients (26.8%) received adjunctive intravitreal corticosteroids. In order to assess the differences between the two therapies, the characteristics of patients receiving intravitreal corticosteroids were determined, as shown in Table 5. Adjunct use of intravitreal corticosteroids was performed in eyes with symptoms of corneal edema (*p* < 0.001) and poorer initial BCVA (logMAR) (*p* = 0.03). Notably, use of intravitreal corticosteroids was associated with a higher likelihood of receiving systemic antibiotic therapy (*p* = 0.048) and a longer systemic antibiotic duration (*p* = 0.03).

### 2.5. Systemic Antibiotic Use

In total, 46 patients (82.1%) received systemic antibiotic therapy, including intravenous and oral antibiotics, either sequentially or in combination. During hospitalization, ceftazidime was the most frequently prescribed regimen (30/46; 65.2%), followed by ceftazidime/levofloxacin (8/46; 17.4%), levofloxacin (4/46; 8.7%), cefazolin (2/46; 4.3%), and norvancomycin (2/46; 4.3%). At the time of discharge, the most frequently prescribed oral regimens were levofloxacin (8/12; 66.7%) and moxifloxacin (4/12; 33.3%). The median total duration of systemic antibiotics (intravenous and/or oral) was three days (interquartile range: 2–5). Univariate analysis demonstrated that favorable visual outcomes were not associated with the use of systemic antibiotics (*p* = 0.37) or its treatment duration (*p* = 0.13) (see Table 1).

## 3. Discussion

This 10-year study (2013–2022), conducted at a tertiary referral center in eastern China, determined the rate of APSE based on a sample of 153,432 cataract surgeries. The observed incidence rate (0.020%; 95% CI: 0.011%–0.029%) was deemed to be comparable with those reported in developed countries of 0.012–0.068% and the rates of 0.01–0.033% reported by other large ophthalmology institutions in China [2,8,15,16,17], but significantly lower than the rate of 0.11% in small- and medium-scale departments of ophthalmology in China [9].

Identification of pathogens via intravitreal specimen cultures is helpful for the diagnosis and choice of antibiotics for endophthalmitis. However, the percentage of positive cultures in the current study was only 32.1%. The culture yield through the vitreous samples (35.4%) and aqueous samples (29.0%) were closer to the lower end of the ranges reported previously for vitreous samples (32.3–88.0%) and aqueous (20.0–71.4%) samples [18,19]. Coagulase-negative *staphylococci* was the most frequently identified organism (12/21; 57.1%), which was also consistent with previous reports [3,12]. Regarding the types of pathogens observed, highly virulent pathogens were detected significantly more commonly in patients with hypopyons and shorter time to onset of symptoms after surgery (median number of days: 1.5 days), in which cases the eyes were deemed to be vulnerable. Patients infected with highly virulent organisms displayed poorer visual outcomes, all having vision below 20/70. Highly virulent bacteria may induce severe inflammation and ocular tissue damage early in the infection process, leaving little chance for improvement. Thus, timely diagnosis is the key to successful APSE treatment.

In the current analysis, initial BCVA (logMAR) and time from onset to initial IVAs showed independent positive correlations with the final BCVA (logMAR), which indicates that the severity of the condition at baseline, as well as untimely treatment, are risk factors for poor vision outcomes. These findings were also in accordance with several previous reports [2,8,17,20]. A better initial BCVA indicates the possibility of a shorter disease course, less damage to intraocular tissues, and a greater chance of a good visual outcome. 

The role of intravitreal corticosteroids in infectious endophthalmitis remains controversial. The ocular inflammatory response caused via bacterial infiltration induces significant damage to retinal tissues. Intravitreal corticosteroids block inflammation by acting on various steps of the inflammatory pathway, including interfering with inflammatory cell congregation, prostaglandin synthesis, and superoxide release [21]. A meta-analysis conducted by Soekamto et al. reported that adjunct use of intravitreal corticosteroids for the treatment of acute bacterial endophthalmitis had no additional visual benefit, regardless of whether the patient underwent PPV [22]. In contrast, a systematic review conducted by Emamiet et al. was inconclusive as to whether the utilization of intravitreal corticosteroids for the treatment of acute endophthalmitis after intraocular surgery is effective due to the lack of sufficient evidence [13] In the current study, the adjunct use of intravitreal corticosteroids may have contributed to a favorable visual outcome in the more severe cases comparable to those not primarily triggering the use of intravitreal corticosteroids. It should be noted that the patients who were treated with intravitreal corticosteroids could have initially had a more serious condition than their counterparts. Therefore, caution should be exercised in drawing a conclusion that adjunctive intravitreal corticosteroids are not beneficial in APSE.

The main foundations of post-operative bacterial endophthalmitis management were derived from the results of the Endophthalmitis Vitrectomy Study (EVS), a prospective randomized controlled trial (RCT) study of 420 patients with bacterial endophthalmitis conducted at more than 20 centers in the United States in the year 1990 [12]. The EVS reported that intravenous antibiotics did not result in a better visual outcome. However, the EVS assessed intravenous forms of amikacin and ceftazidime, both of which have poor intravitreal penetration [23], in contrast to newer drugs such as fluoroquinolones [24]. Hooper et al. reported that the adjunctive use of moxifloxacin can exert a beneficial impact on the final visual outcome in APSE [25]. Grzybowski et al. suggested that certain patients with severe disease may benefit on a case-by-case basis, but the elevated cost of systemic agents and possible related toxicity remain important considerations [14]. In the current study, patients who received systemic antibiotics exhibited similar visual outcomes as compared to those who did not. Thus, it remains inconclusive as to whether this treatment is unnecessary, as the use of adjunctive systemic antibiotics was likely driven by the initial severity of disease. Given limited evidence from relevant randomized clinical trials, further studies on the role of systemic antibiotics, antibiotic selection, and the treatment duration of APSE are needed to provide guidance for optimal antibiotic management of these infections.

The current study had several limitations. First, the retrospective design may have led to bias regarding the nature of data collected, with a substantial individual variability in follow-up time and treatment regimen. The treatment strategy for each case of APSE was based on the clinical judgment of the treating physician. Due to the low incidence of APSE, designing RCTs is difficult. However, this series represents real-world clinical data, and reflects the complex and varied treatment strategies of APSE that clinicians prescribe in practice, adding to the existing knowledge of its management. Second, the rate of positive culture of the vitreous samples was relatively low. This may have been caused by delays in sample plating or in the transfer to the laboratory for culture, which would reduce the viability of the organisms. Several publications have reported recent advances in technology for the microbial diagnosis of endophthalmitis, including nanopore amplicon sequencing, blood culture bottles, and PCR detection [26,27], which could be used in the diagnosis of bacterial endophthalmitis in our hospital. Finally, this was a single-center study, and therefore our findings may not be generalizable to other hospitals with different patient populations or local microbiology. However, considering that our hospital is a representative ophthalmic center in eastern China performing about 15,000 cataract surgeries per year, the results of the current study are representative of outcomes in large-scale referral centers in China.

## 4. Materials and Methods

### 4.1. Study Population

This study is a retrospective, non-comparative case series. The Fudan University Eye and Ear, Nose, and Throat (FUEENT) Hospital is a large university teaching hospital that accepts referrals of patients with endophthalmitis from the eastern provinces of China. The medical histories of all patients diagnosed with APSE from January 2013 to December 2022 were retrospectively reviewed. The inclusion criteria were as follows: clinical or laboratory-diagnosed infectious endophthalmitis occurred within 42 days after phacoemulsification or small-incision extracapsular extraction combined with primary intravitreal lens implantation. The exclusion criteria were as follows: cataracts in conjunction with other intraocular surgery (including anti-glaucoma surgery, corneal transplantation, and vitrectomy); patients with suspected endogenous endophthalmitis; or post-traumatic endophthalmitis (Figure 1). Endophthalmitis was clinically diagnosed and managed under a standardized protocol (the ninth revision of the *International Classification of Diseases*, code 360.0). 

### 4.2. Treatment

All patients underwent an initial antibiotic therapy comprising a combination of broad-spectrum topical agents and IVAs, followed by specific antibiotics according to the culture results. The initial medication consisted of IVAs (norvancomycin 0.8 mg/0.1 mL and/or ceftazidime 2.25 mg/0.1 mL) and intensive topical antibiotics (tobramycin 0.3%/gatifloxacin 0.3%/levofloxacin 0.5%/moxifloxacin 0.5%). A second intravitreal injection of the same antibiotics was administered at 48 h if there was no improvement in clinical symptoms or signs. Indications for PPV followed the EVS guidelines [12]. In addition, patients may have also undergone PPV if the surgeon determined that this was in the patient’s best interest. The systemic antibiotics (intravenous and/or oral) used included norvancomycin, ceftazidime, cephazolin, levofloxacin, and moxifloxacin. Intravitreal dexamethasone (0.4 mg/0.1 mL) or triamcinolone (1 mg/0.1 mL) were administered based on the clinical judgment of the treating physician.

### 4.3. Vitreous/Aqueous Sample Collection and Culture

Undiluted vitreous and aqueous samples were obtained, via either needle aspiration or PPV at the beginning of the surgery and injection in all cases, and the specimens were sent to the Microbiology Department of the FUEENT Hospital for direct smears and cultivation. A 30 G needle was used to enter the anterior chamber through a limbal incision, and 0.1–0.2 mL aqueous humor was extracted. Vitrectomy specimens, ranging from 1 mL to 1.5 mL, were collected using a vitreous cutter connected to a 2 mL sterile syringe with manual suction. For the tap-and-inject approach, vitreous fluid (0.1–0.2 mL) was acquired using a 23 G needle. For the isolation of fungi, the specimens were inoculated on Sabouraud’s dextrose agar and incubated at 27 °C for 7–14 d. For aerobic bacteria culture, the specimens were inoculated into brain heart infusion broth and incubated at 37 °C for 7 d. The broth was then sub-cultured onto blood agar plates and incubated for 48 h. For anaerobic culture, the specimen was inoculated onto blood agar and loaded into a sealed bag together with a commercial agent, which absorbed oxygen to produce carbon dioxide, and then incubated at 37 °C for 5–7 d. A MicroScan AutoScan system (Dade MicroScan Inc., Sacramento, CA, USA) was employed for bacterial identification.

### 4.4. Data Collection

The data collected from patients with APSE included: demographic characteristics (age, gender, and systemic disease), onset year of endophthalmitis, time interval from cataract surgery to onset of endophthalmitis, time from onset of endophthalmitis to IVAs, microbiology culture results, presence of hypopyons, initial treatment, presenting BCVA, final BCVA, and the follow-up duration. Data on adjunctive treatment modalities, including intravitreal corticosteroids and systemic antibiotics, were also collected. The definition for high virulence was based on the associated ocular damage and was consistent with other published reports [28,29,30]. Highly virulent microorganisms isolated in this study included *Staphylococcus aureus*, *Bacillus*, *Streptococcus*, *Pseudomonas aeruginosa,* and *Escherichia coli*. Coagulase-negative *staphylococci* was considered as having low virulence.

### 4.5. Statistical Analysis

All analyses were carried out using SPSS software (version 22.0; IBM Corp, Armonk, NY, USA). Categorical variables were summarized using proportions and analyzed using the Fisher’s exact test or Chi-square test. Continuous variables were summarized using medians with interquartile ranges as appropriate according to normality and analyzed using the Mann–Whitney U test. According to the standards for the characterization of vision loss updated by the World Health Organization in 2003, the criteria for favorable vision was set as BCVA ≥ 20/70 [2]. Variables with *p*-values less than 0.1 in the univariate analysis were included in the multiple linear regression models. Snellen vision was converted into the logMAR. A two-tailed *p*-value <0.05 was considered statistically significant.

## 5. Conclusions

In conclusion, this study revealed that three factors (initial BCVA, time from onset to initial IVAs, and positive culture of highly virulent pathogens) showed independent positive correlations with the final BCVA. Patients infected with highly virulent organisms had poor visual outcomes, all having vision below 20/70. This finding underscores the need to optimize measures to prevent infection and to optimize techniques for early detection of infection and identification of causative organisms.

## Figures and Tables

**Figure 1 antibiotics-12-01670-f001:**
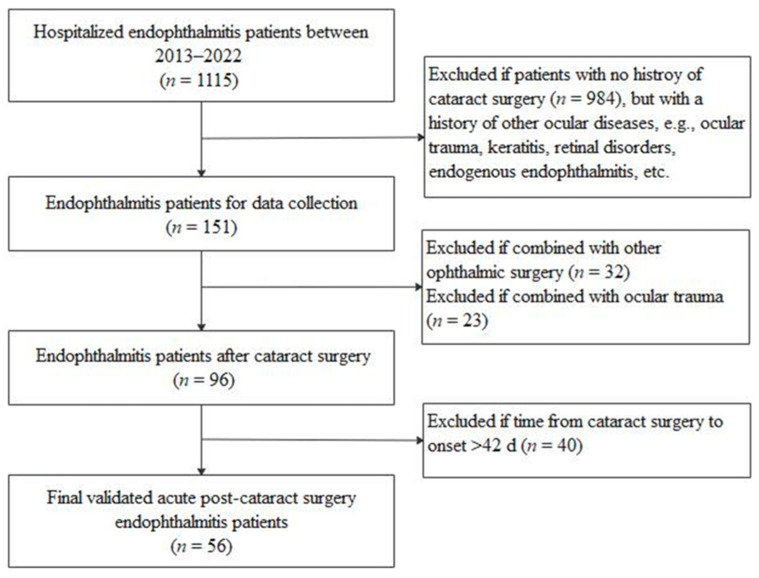
Flowchart of the recruitment of acute post-cataract surgery endophthalmitis patients.

**Table 1 antibiotics-12-01670-t001:** Demographics, clinical characteristics, and management of acute post-cataract surgery endophthalmitis cases.

Variable	Total (*n* = 56)	Final BCVA ≥ 20/70	Final BCVA < 20/70	*p*-Value
		(*n* = 21)	(*n* = 35)	
Age (years)	70 (59, 74)	63 (54, 75)	71 (60, 73)	0.63
Sex/male	27 (48.2%)	12 (57.1%)	15 (42.8%)	0.3
Left eye	25 (44.6%)	12 (57.1%)	13 (37.1%)	0.14
Time from cataract surgery to onset of endophthalmitis (days)	4 (1, 11)	7 (3, 16)	3 (1, 8)	0.008
Time from onset to initial IVAs (days)	3 (2, 7)	2 (2, 4)	5 (2, 9)	0.09
Diabetes mellitus	10 (17.8%)	2 (9.5%)	8 (22.8%)	0.21
Presence of corneal edema	30 (53.6%)	10 (47.6%)	20 (57.1%)	0.49
Presence of hypopyon	20 (35.7%)	6 (28.6%)	14 (40.0%)	0.39
Positive culture	18 (32.1%)	5 (23.8%)	13 (37.1%)	0.3
High virulence of pathogens	8 (14.3%)	0 (0.0%)	8 (22.8%)	0.02
Initial treatment with PPV + IVAs	44 (78.6%)	16 (76.2%)	28 (80.0%)	0.74
Use of intravitreal corticosteroids	15 (26.8%)	6 (28.6%)	9 (25.7%)	0.82
Use of systematic antibiotics	46 (82.1%)	16 (76.2%)	30 (85.7%)	0.37
Total systemic antibiotic duration (days)	3 (2, 5)	3 (1, 4)	3 (2, 7)	0.13
Initial BCVA (logMAR)	2.3 (1.8, 2.7)	2 (1.0, 2.3)	2.3 (2, 2.7)	0.03

IVAs = intravitreal antibiotics; PPV = pars plana vitrectomy; and BCVA = best-corrected visual acuity.

**Table 2 antibiotics-12-01670-t002:** Multivariate linear regression for predicting final visual outcome.

Variable	Standardized *β*	*p*-Value
Time from cataract surgery to onset of endophthalmitis (days)	−0.078	0.506
Time from onset to initial IVAs (days)	0.383	<0.001
Infected with highly virulent pathogens	0.286	0.018
Initial BCVA (logMAR)	0.407	<0.001

IVAs = intravitreal antibiotics; and BCVA = best-corrected visual acuity.

**Table 3 antibiotics-12-01670-t003:** Microbiological profile of patients with acute post-cataract surgery endophthalmitis.

	Pathogens from the Aqueous	Pathogens from the Vitreous
Total	31	48
Negative	22	33
Coagulase-negative *staphylococci*	6	7
*Staphylococcus aureus*	1	1
*Streptococcus dysgalactiae*	0	1
*Bacillus cereus*	0	1
*Pseudomonas aeruginosa*	1	2
*Escherichia coli*	1	1
*Aspergillus* + Coagulase-negative *staphylococci*	0	1
α-hemolytic streptococci + Coagulase-negative *staphylococci*	0	1

**Table 4 antibiotics-12-01670-t004:** Characteristics of patients grouped via bacterial types.

Variable	Highly Virulent Pathogen (*n* = 8)	Less Virulent Pathogen (*n* = 48)	*p*-Value
Age (years)	70 (57, 74)	69 (62, 77)	0.61
Sex/male	3 (37.5%)	24 (50.0%)	0.79
Left eye	2 (25.0%)	23 (47.9%)	0.41
Time from cataract surgery to onset of endophthalmitis (days)	1.5 (1, 4)	5 (2, 13)	0.04
Time from onset to initial IVAs (days)	1 (0.25, 2)	4 (2, 7)	<0.001
Diabetes mellitus	2 (25.0%)	8 (16.7%)	0.94
Presence of corneal edema	6 (75.0%)	24 (50.0%)	0.35
Presence of hypopyon	6 (75.0%)	14 (29.2%)	0.04
Initial BCVA (logMAR)	2.5 (2.3, 2.7)	2.3 (1.5, 2.7)	0.08
Final BCVA (logMAR)	2.5 (1.0, 3.0)	0.9 (0.3, 2.3)	0.02
Final BCVA ≥ 20/70	0 (0.0%)	21 (43.8%)	0.02

IVAs = intravitreal antibiotics; and BCVA = best-corrected visual acuity.

**Table 5 antibiotics-12-01670-t005:** Characteristics of patients receiving intravitreal corticosteroids.

Category	Corticosteroids Group (*n* = 15)	Non-Corticosteroids Group (*n* = 41)	*p*-Value
Age (years)	70 (53, 74)	68 (59, 74)	0.89
Sex/male	9 (60.0%)	18 (43.9%)	0.29
Left eye	9 (60.0%)	16 (39.0%)	0.16
Time from cataract surgery to onset of endophthalmitis (days)	2 (1, 7)	5 (2, 12)	0.11
Time from onset to initial IVAs (days)	2 (1, 7)	4 (2, 7)	0.33
Diabetes mellitus	4 (26.7%)	6 (14.6%)	0.52
Presence of corneal edema	14 (93.3%)	16 (39.0%)	<0.001
Presence of hypopyon	8 (53.3%)	12 (29.3%)	0.10
Positive culture	4 (26.7%)	14 (34.2%)	0.84
Highly virulent pathogen	2 (13.3%)	6 (14.6%)	1.00
Initial treatment with PPV + IVAs	13 (86.7%)	31 (85.4%)	0.37
Use of systematic antibiotics	15 (100.0%)	31 (75.6%)	0.048
Total systemic antibiotic duration (days)	5 (3, 7)	3. (0.5, 4.5)	0.03
Initial BCVA (logMAR)	2.7 (2.3, 2.7)	2.3 (1.6, 2.3)	0.03
Final BCVA (logMAR)	1.3 (0.3, 2.3)	1.1 (0.4, 2.3)	0.56
Final BCVA ≥ 20/70	6 (40.0%)	15 (36.6%)	0.82

IVAs = intravitreal antibiotics; PPV = pars plana vitrectomy; and BCVA = best-corrected visual acuity.

## Data Availability

All data generated or analyzed during this study are included in this article. Further inquiries can be directed to the corresponding author.

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
