# Peer review of "Management and Prognosis of Acute Post-Cataract Surgery Endophthalmitis: A 10-Year Retrospective Analysis in Eastern China"

_antibiotics, 2023, doi:10.3390/antibiotics12121670_

Round 1

Reviewer 1 Report

Comments and Suggestions for Authors

Even if the paper is  really well done, there's no new information in it, and I don't consider that it could be published, Results and conclusions are already known.

Reviewer 2 Report

Comments and Suggestions for Authors

The well-written manuscript entitled « Management and Prognosis of Acute Post-Cataract Surgery Endophthalmitis: a 10-Year Retrospective Analysis in Eastern China » provides work up of a ten-year series of post-cataract surgery endophthalmitis in a single center in Eastern China. The major concern with this manuscript, an inherent bias associated with the retrospective design of this series is that there did not exist a defined therapeutic standard, so that treatment aggressvity was driven by the severity of initial clinical presentation, as correctly outlined by the authors (lines 200-3). In consequence, the role of systemic antibioticas and intravitreal corticosteroids,both of which were used in the more aggressive instances did not show a benefit, as the authors conclude, while they may well have prevented a relevantly worse outcome. The conclusion « adjunctive intravitreal corticosteroids or systemic antibiotics should be used with caution » is thus not supported.

Further comments :

·       All patients had undergone phako or small incision ECCE. What about complications? Were they excluded (where are they represented in figure 1 (combining with other surgery)? Complicated ctararact surgery should be included in the multivariate regression analysis.

·       Please re-consider the discussion and conclusions based on the case selection and bias introduced by tailoring treatment to the severity of disease at presentation

·       Surgical trauma may be linked to the outcomes. Therefore, duration of surgery has to be recorded and to be included in multivariate analysis.

·       There exists a mis-understanding of the correlation of time to IVA and high virulence of pathogens. This correlation is indirect in that a shorter time to IVA and more virulent germs were associated with poorer outcomes.

·       A definition for high virulence is lacking. It must be clear, if this is based on microbiological characteristcs or the associated ocular damage. In the latter case, staph aureus, pseudomonas and all endotoxin producing bacteria would have to be categorized as highly virulent.

·       Was a microbiological sample taken in any case ? I assume, that undiluted samples of aqueous and vitreous were collected prior to any antibiotic use also in referral cases ?

·       The relatively low microbial growth rate is worth assessing the underlying processes to improve the test performance (for example time to culture). It remains to be explained, why – given the low test sensitivity, ueubacterial PCR was not routinely applied.

·       The total number of tested antibiotics (table 5) i slow. What argued against testing all antibiotics in clinical use. Obviously in the absence of a standard of care, many different first-line antibiotics were used, the most frequent being Ceftazidime, which was tested in just 3 instances! Where ist he rationale?

·       Given the unsystematic testing of antiobiotic resistence, no conclusions are supported by these data, the percentages in lines 105-9 are misleading/wrong, would recommend to remove the whole paragraph along with the misleading tzable 5. If the latter should be maintaines, the column with percentage is to be removed.

·       How often was the antibiotic regimen asdopted according to the outcomes of cultures and resistence testing (lines 146-7)?

Minor points :

·       In line 23, time from surgery to onset of endophthalmitis should be detailed for patients with better and worse outcomes Instead of the median of the whole group as displayed in table 1.

·       In figure 1, 984 cases were excluded because of no history of cataract surgery, these also did not have combined surgery and or late onset endophthalmitis. For the interested reader, it wopuld be highly interesting to learn more about this group in a table with the diagnoses. They may well represent a cohort deserving reporting in an independent paper.

·       Why were combined surgeries (n=32) excluded ?

·       One eye needed evisceration/enucleation : What was the underlying microbiological growth?

·       Lines 159-60 : probably true, but not supported by the data presented here.

·       Lines 172-7 : This information has to be differentiated. The role of IVA in case of immediate vitrectomy (the more severe presentations) differs from that receiving a tap-and-inject protocol. The same accounts for intravitreal CS.

·       Line 180-1 : The use of CS may have contributed to an as favourable outcome in the more severe cases compared to that not primarily triggering the use of CS.

·       As avitreoretinal surgeon, the sentence (line 183) « This may be the result of clinicians being concerned about the spread of infection » did not more than trigger a smile to me. The true surgical concern is to do anything to escape a deleterious outcome (enucleation) in ocular disease in severe primary presentation.

·       Line 203 : The widely heterogenous surgical and medical strategy does not argue in favour of strong adherence to the EVS guidelines. Would therefore remove this sentence or show, how these were systematically applied.

·       Line 212: replace by should be considered as adjunct… instead of alternative.

·       Line 252 : True also for tap-and-inject ?

I’d like to thank the authors to allow me to add to improvement, of their manuscript which is worth being done.

Comments on the Quality of English Language

minor changes, for example remove significant if p values are displayed, replace symptoms by presence (line 102), remove only (line 76), staphylococci is pluralis, change was to were (line 151) 

Reviewer 3 Report

Comments and Suggestions for Authors

This study provides an overview of the incidence, clinical presentation, visual outcomes, microbiological features, and management methods of all APSE cases in a large university teaching hospital and aimed to explore the effectiveness of adjunctive treatment for APSE and to evaluate potential predictors for visual outcomes. There are 2 questions should be clarified.

Because twenty-nine underwent cataract surgery at authors’ hospital and the other 27 underwent cataract surgery at outside hospitals. My question is how to calculate the overall APSE rate from different sources (hospitals)?

Dis “Time from onset to initial IVA” include all cases? --> 44 (78.6%) and 12 (21.4%) eyes were initially treated with PPV+IVA and IVA, respectively?

Reviewer 4 Report

Comments and Suggestions for Authors

This retrospective, mono-centric study provides an overview of the incidence, clinical presentation, visual outcomes, microbiological features, and management methods of all 56 APSE cases (out of 153.432 cataract surgeries) in a large university teaching hospital in eastern China over ten years.  The study also explored the effectiveness of adjunctive treatment for APSE (intravitreal steroids and systemic antibiotics should be used with caution) and evaluated potential predictors for favorable visual outcomes (initial BCVA, time from onset to initial IVA, virulence of pathogens).

In summary, this is a clinically very relevant and interesting to read manuscript dealing with an emergency situation after cataract surgery, perfectly written.

Comments:

Line 71: ---PPV + IVA and IVA…is PPV + IVA not sufficient? Or: …PPV + IVA or IVA alone?

Line 76: …only 1 eye was enucleated or eviscerated….please change to: only 1 eye was enucleated and no eye was eviscerated.

Reviewer 5 Report

Comments and Suggestions for Authors

The author propose a paper on Management and Prognosis of Acute Post-Cataract Surgery Endophthalmitis: a 10-Year Retrospective Analysis in Eastern China

Endophthalmitis are one of the most intricate issue in infectious disease of ocular surgery and its incidence is widely different coutry by country.

The paer has a merit to propose a very interesting number of cases and a significant data on local epidemiology

I have some minor criticisms

Please provied a possible better  explaination on why your results are is comparable with those reported in developed countries but but significantly lower than the occurrence rate of 0.11% in small- and medium-scale departments of ophthalmology in China.

Please propose why staph are most frequent and if you use nasal swab for staph in your pre surgical screening

Round 2

Reviewer 2 Report

Comments and Suggestions for Authors

would recommend to remove the text in lines 139-44, which is not relevant, but throws its shadows onto the colleagues from smaller units.

Comments on the Quality of English Language

The linguistic quality has not improved with the revision. 

Author Response

Thank you very much for taking the time to review this manuscript. Please find the detailed responses below and the corresponding revisions/corrections highlighted/in track changes in the re-submitted files。

Comment 1: I would recommend to remove the text in lines 139-44, which is not relevant, but throws its shadows onto the colleagues from smaller units.

Response to comment 1: Thank you for your valuable comments. We have removed the text in lines 139-44 according to your comments.

Comment 2: The linguistic quality has not improved with the revision.

Response to comment 2: Thank you for your comments. The manuscript has now been thoroughly revised by a native English speaker.